# How swarming bats can use the collective soundscape for obstacle avoidance

**Dieter Vanderelst** [ID][1]*, **Herbert Peremans**[2]

**1** Department of Biological Sciences, University of Cincinnati, Cincinnati, Ohio, United States of America,
**2** Department of Engineering Management, University of Antwerp, Antwerp, Belgium

\* vanderdt@ucmail.uc.edu

**Data availability statement:** The data and the code (including documentation) is available at https://osf.io/qsm8c/. The DOI is https://doi.org/10.17605/OSF.IO/QSM8C.

## Abstract

Some echolocating bats, such as *Tadarida brasiliensis*, fly in groups when emerging from or entering caves. In large, dense swarms, distinguishing self-generated echoes from the multitude of calls and echoes produced by others presents a significant challenge – akin to a cocktail party nightmare. While spectral jamming responses have been proposed as a solution, this mechanism is unlikely to be effective in such conditions. Here, we propose an alternative hypothesis: rather than isolating their own echoes, bats might navigate by relying on the local amplitude gradient of the collective soundscape. To test this, we developed an agent-based simulation of bats flying through corridors, demonstrating that they can avoid obstacles, including other bats and corridor walls, without distinguishing individual echoes. Our findings suggest that in dense swarms, bats can exploit the emergent acoustic environment to maintain safe distances. The current paper also suggests shifting the perspective on jamming itself. Rather than framing overlapping signals solely as a source of interference, our findings highlight that these signals can also carry useful information, reframing the problem from conflict to cooperative signal processing.

## Author summary

Some echolocating bats fly in dense groups when emerging from or returning to their roosts. These groups create an overwhelming soundscape of calls and echoes, probably making it impossible for individual bats to identify their own echoes, a challenge known as the "cocktail party problem." Despite this, these bats successfully avoid collisions with each other and with their surroundings. How do they achieve this?

Using computer simulations, we explored a potential explanation: bats may not need to isolate their own echoes in such situations. Instead, they could use the overall loudness of the soundscape created by the group to steer away from obstacles. Our simulations show that this simple strategy allows virtual bats to navigate corridors and avoid collisions, even in dense swarms.

**Funding:** This work was funded by NSF (National Science Foundation, https://www.nsf.gov) grant IOS-2034885 to DV. The funders did not play any role in the study design, data collection and analysis, decision to publish, or preparation of the manuscript.

This finding reframes the idea of "jamming" from overlapping signals being solely a source of interference to them being a shared resource that bats can exploit. Our study suggests that swarming bats may use the collective soundscape as a form of cooperative signal processing, offering a new perspective on how these animals navigate their (acoustic) world.

## Introduction

Bat echolocation is an active sense that stimulates the environment by injecting acoustic energy, bat calls, and processes the environment's response, sonar echoes. This allows the animal to adapt its vocalizations to optimize sensory input for perception [1]. However, a limitation of this mode of sensing arises when multiple individuals echolocate in the same space. In this case, calls and the resulting echoes generated by conspecifics might cause jamming [2]. Jamming can occur when the echoes are masked by the signals of other bats (i.e., a *detection* problem). A more critical aspect of jamming is its potential to interfere with the ability of the bat to recognize its own echoes among the multitude of calls and echoes (i.e., a *selection* problem) generated by others. It is assumed that bats need to be able to identify their echoes among other signals, such as calls and echoes generated by others, to be able to extract timing and spectral cues and, therefore, to echolocate successfully, e.g., [3–5].

The species that potentially suffers the most from jamming is the Brazilian free-tailed bat (*Tadarida brasiliensis*). This is one of the most abundant bat species in the western hemisphere [6]. The maternal colonies in caves can contain up to a million animals [7, 8] and are some of the largest known animal aggregations in the world [9]. Unlike many other bat species, where the emergence patterns are diffuse and show little group integrity, large colonies of *T. brasiliensis* often (but not always [10]) emerge in a tight serpentine column [7,10]. Emergence rates can vary from 100 to more than 30,000/min [7,8].

Dense groups of bats emerging from a cave while echolocating [8,11,12] result in a cocktail party nightmare [7,12]. A bat's emission will be reflected by many other individuals as well as by cave walls. Bats also pick up emissions from others and some of the echoes resulting from these calls. A similar scenario might occur when these bats reenter the roost while echolocating [13].

There is currently no specific hypothesis on how these bats use echolocation to avoid colliding with walls and other bats in those circumstances. In other species of bats, studies have found that bats flying in groups alter their echolocation signals, a phenomenon referred to as a spectral jamming avoidance response (JAR). The spectral JAR has been hypothesized to help mitigate jamming by reducing the overlap between echolocation signals and masking signals, thereby improving the detection and selection of relevant echoes. Studies reporting such frequency shifts in bats hunting in proximity to each other suggest that this response could be an adaptive strategy to enhance prey detection and localization when conspecifics are present, for example, [2,14,15]. Frequency shifts have also been reported in bats exposed to playback of overlapping signals [16,17]. However, the ubiquity of JAR in bats remains debated. Several studies have found no JAR in bats [3,5,18,19].

Although *T. brasiliensis* could alter their calls and exhibit a spectral jamming response [11,20], this is unlikely to work when many bats fly together [12]. However, critically, unlike other species of bats studied when flying in groups, *T. brasiliensis* (and other species) emerging from roosts do not hunt and locate prey. These animals are engaged in a different task, that is, collision avoidance. In this paper, we advance an alternative hypothesis to explain how

*T. brasiliensis* (and potentially other swarms of bats) could cope with the cocktail party nightmare while not engaged in foraging. We suggest that these bats do not need to detect or select their echoes among the multitude of acoustic inputs. Instead, bats could use the local amplitude of the acoustic field generated by calls and echoes to steer away from obstacles. We test this hypothesis by creating an agent-based simulation of bats flying in passageways and exploring whether they can negotiate obstacles, i.e., other bats and the corridor walls, without being able to distinguish their echoes from the multitude of acoustic inputs they receive when flying in dense groups.

## Simulation setup

The simulation method is discussed in detail in the Methods section. In the following, we provide an overview of the simulation approach before discussing the results.

We created simulated arenas with walls consisting of point reflectors spaced at 0.1 m intervals in the horizontal plane and uniformly scattered in the vertical plane ($\pm$ 1 m). Each arena consisted of 5 m wide corridors. For each arena, we defined a cylindrical starting region in which all agents were spawned at the start of the simulation. We also defined one or two cylindrical exit regions. When an agent reached an exit region, we removed it from the simulation to model the bat leaving the cave or corridor. In the simulations reported, the agents start at different heights but do not change their flight height, that is, their position $z$ is fixed. We did this to simplify the simulations, but also because bats, including *T. brasiliensis* [21], following a flight path, have been observed to maintain a constant flight height [22].

We determined the number of agents in our simulations ($n = 25$) based on the density of bats observed by Weesner et al. [52] for *T. brasiliensis* emerging from a cave (see Methods for details). We opted for this moderate number of agents because the computational complexity of the simulations increases non-linearly with the number of simulated agents. To ensure that the simulations remained computationally feasible while still maintaining ecological realism, we utilized a smaller number of agents that preserved a realistic density rather than scaling up the number of agents to match larger group sizes. To ensure that the results also scale to simulations with more bats, we include results on runs with 50 agents, scaling the arena to keep the density constant.

Each simulation was run until at least 50% of the agents had reached the end region and were removed from the simulation. For each step of the simulation, we used the sonar equation [23] and the directionality of the bat *Phyllostomus discolor*'s emission and hearing as simulated by Vanderelst et al. [53] to calculate the perceived amplitude in decibels of several acoustic inputs for each agent. We used the hearing and emission directionality averaged from 30 to 50 kHz, as this is the frequency range of calls used by *T. brasiliensis* as they emerge from caves [24].

The directionality of the *Phyllostomus discolor* seems fairly typical for bats. Jakobsen et al. [54] measured the emission beamwidths of six vespertilionid bat species in a flight room and found that their half-amplitude angles (-6 dB) were approximately $\pm 37°$. The simulated beamwidth of *Phyllostomus discolor* at the frequencies used in this study is somewhat larger, at about $\pm 50°$ in both azimuth and elevation. Regarding hearing directionality, the opening angle of the main lobe in the simulated head-related transfer function for *Phyllostomus discolor* spans approximately 70° in azimuth, which aligns with the main lobe extents reported by Obrist [55] for several bat species. Similarly, this extent is comparable to the hearing directionality reported for *Eptesicus fuscus* by Aytekin et al. [25]. In summary, the hearing and emission directionality used in this paper is not higher than what has been observed in

other bats. Therefore, it is unlikely that unusually high directionality in emission or hearing enables our agents to filter out or suppress spurious acoustic input.

We calculated the perceived loudness of the calls of other agents (Eq 1), taking into account emission and hearing directionality. We also calculated the sound level in decibels for echoes that returned from the bodies (Eq 2) of other agents [26] and the arena's walls (Eq 3) as a result of an agent's emission. Finally, we simulated the loudness of the secondary body echoes (Eq 4) and the echoes of the walls (Eq 5). These are the echoes received by an agent due to other agents' calls. Therefore, we calculated the loudness of five types of acoustic input for each agent at each step of the simulation. As detailed in the Methods section, the acoustic inputs received by each agent in its left and right ear were converted to an impulse response representation. Next, this representation was convolved with an artificial bat-like call (50 to 30 kHz, 9 ms long [7]) to obtain the waveform picked up by each ear. The waveform was then processed using a bat cochlea model [27]. Cochlear responses for the left and right ears were calculated for the full interpulse interval. However, our agents only used the first half of the calculated cochlear responses. This is a simple approach to model that biological echolocators require time to process the acoustic input.

The controller for each agent was simple and is based on our previous results demonstrating that interaural level differences could be used to guide simulated and robotic bat models away from obstacles [28,29]. To calculate the interaural level difference, we processed the (first half of the) cochlear model output for the left and right ear. We looked for the time of occurrence of the maximum in the simulated cochlear responses for the left and right ears. Next, we integrated the cochlear output in a window of 1 ms around this time point in the left and right ear, i.e., akin to the approach used in [28,29]. The resulting values were subtracted from each other to obtain the interaural level difference. The agent turned to the left or right, depending on the sign of the interaural level difference. If the left (right) ear received the loudest input, the agent turned right (left).

It is important to note that the simulated agents do not distinguish between the different acoustic inputs. In other words, the agents did not identify whether the part of the cochlear output used to calculate the interaural level difference included a self-generated echo or an echo or call generated by another agent. Moreover, in the current study, we do not use the distance or time of arrival of the echoes because we assume that the bat cannot differentiate its echoes from other acoustic inputs it encounters in a swarm.

Unfortunately, there is a lack of data on the echolocation and flight behavior of bats navigating caves. Specifically, to the best of our knowledge, there are no measurements available for how often *T. brasiliensis* or any other bat species calls while negotiating caves. Lin et al. [30] report on call rates of groups of *Miniopterus fuliginosus* leaving a cave. However, as these authors also acknowledge, their measurements are likely inflated by additional reflections from the cave walls being counted as calls. However, their data suggest that bats increase their call rates as the number of bats flying concurrently increases.

In the absence of direct data on the call rates used in caves, we use data from [31] to parametrize the interpulse interval in our model. These authors studied *T. brasiliensis* flying in a flight room. In their experiments, bats flew at speeds of 4–5 m/s (substantially slower than the speed we modeled), calling 15 to over 40 times per second depending on environmental complexity and social context. Higher call rates (up to 40 Hz) were achieved through call grouping. In the presence of other (simulated or real) bats, the bats used an intermediate call rate of approximately 25 Hz. We use this intermediate rate as it strikes a balance between the maximum and minimum rates observed under different conditions and serves as a reasonable estimate for our model. The agents were assumed to call asynchronously. To model this, the

arrival times of the acoustic inputs for a given agent that arose from others' calls (i.e., others' calls and the resulting wall echoes) were randomly shifted in time.

The flight speed of *T. brasiliensis* inside caves remains unmeasured. In the absence of direct data, our model was based on flight speeds observed as *T. brasiliensis* entered [21] or exited [13,32] caves. Based on this data, agents were set to a flight speed of 9 m/s. However, if an agent collided with the walls, it was rotated to face the center line of the corridor, and its speed was reduced to 1 m/s. The agent then reaccelerated until it reached 9 m/s. Wall collisions were registered when an agent ventured outside the arena. It should be noted that *T. brasiliensis* significantly slows down as it approaches cave openings [13] and speeds up while leaving caves [21]. This suggests that, inside caves, *T. brasiliensis* might fly slower than 9 m/s, and the flight speed modeled here might be an upper limit. The magnitude of an agent's rotational velocity (in degrees per second) was determined based on the current speed of the agent, limiting the g force to 2, based on the maximum turning rates observed for several species of bats by Holderied [33]. After we determined the rotational and linear velocity of each agent for the current simulation step, we displaced the agents by applying the appropriate rotation and translation.

We conducted simulations in three corridors to assess the efficacy of the control algorithm. All corridors were about 5 m wide. The first corridor was branched, beginning as a single curved corridor and then dividing into two after about 30 meters. The second corridor was curved, forcing the agents to turn left and right. The third corridor started at 5 meters wide, widened after about 10 meters, and then narrowed back to 5 meters wide. Here, we only report in detail on the branched corridor, as the results for the other two were comparable. The results for the other two corridors are included in S1 File (Figs B and D). When testing whether the presented control algorithm scales to larger groups of simulated agents, we use a scaled-up version of the branched arena.

## Results

The code and data for this paper are available at https://osf.io/qsm8c/.

### Default settings

We simulated 15 runs in the branched, curved, and widened arenas. Fig 1A shows the result of one simulation run in the branched arena. The visualization shows that the agents could negotiate the curve and the fork in the corridor and reach the exit of the arena using only the difference in the strength of the simulated acoustic input in the left and right ears. Fig 1B, we also visualize the source of the loudest acoustic input for each agent at each simulation step. This visualization is based on the strength of the acoustic inputs before converting them into a waveform. Once the acoustic inputs are converted to a waveform, information about the loudness of individual echoes or calls is no longer available.

Fig 1A and 1B show that the agents responded to the arena's walls as the paths followed their curvature and the group separated at the point of the fork. However, this figure does not make it clear that the agents also avoided each other. To demonstrate this, we ran the same simulations but turned off the echoes returning from other agents' bodies and their calls. Under these conditions, the agents only use the echoes that return from the walls (including those generated by other agents). It should be noted that these simulations were not aimed at modeling bats. We include this simulation variation only to show that the agents did indeed avoid each other in Fig 1A and 1B using calls emitted by others and echoes from their bodies. Fig 1C shows that under these conditions, agents mostly converge on a path in the middle of the corridor. This shows that the agents shown in Fig 1A and 1B responded to each other by

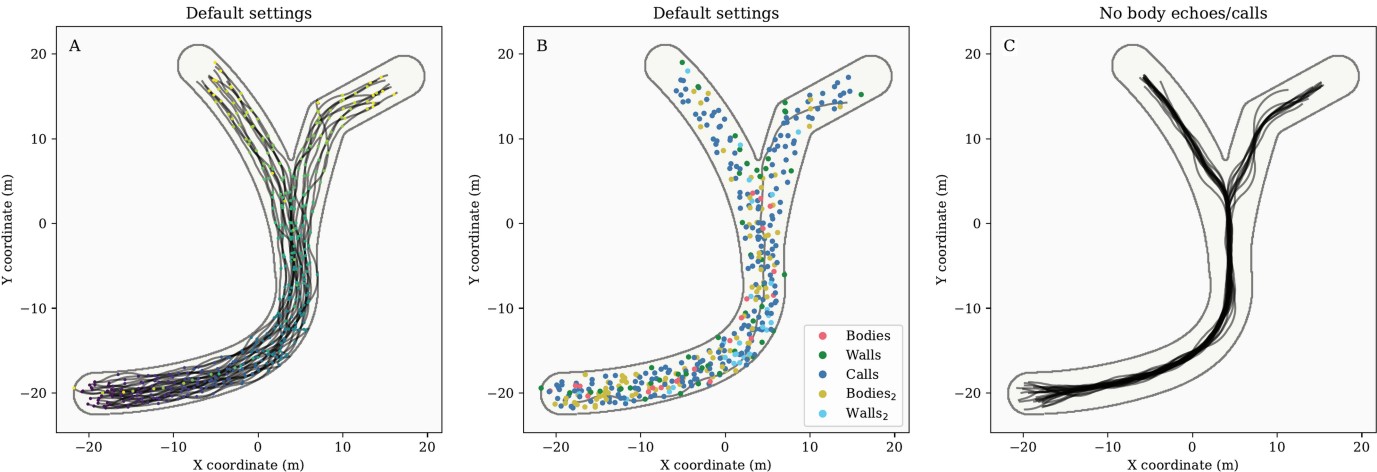

**Fig 1. Examples of the paths of the simulated agents in the branched arena (1 run out of the 15 iterations).** The start location of each agent is depicted by a black dot. (A-B) Example results for the default settings. (A) The simulation step count is visualized for every 10 steps using a color scale to indicate the simulation step number. Similarly colored dots indicate the positions of the agents during the same iteration of the simulation. (B) The color of the dot shows which type of acoustic input was the loudest at each plotted position for each agent, i.e., the dots correspond to the positions and loudest echoes for each agent. We also plotted the path of a single selected agent. The dots on this line correspond to the loudest input received by this agent. (C) Example results for a simulation in which echoes from others' bodies and others' calls were disabled.

spreading across the width of the corridor. This behavior can also be observed in the curved and widening arenas (Figs A and C in S1 File), confirming that the agents used calls emitted by others and echoes from their bodies to avoid each other.

We evaluated the agents' ability to avoid walls and other agents using two metrics. First, we tracked the number of wall collisions, defined as instances where agents attempted to cross the arena boundary. When an agent crossed the boundary, it was repositioned at the crossing point and reoriented to face the arena's centerline. Second, we analyzed the distribution of the shortest distance between an agent and walls or other agents. Specifically, we recorded the proportion of simulation steps during which agents were within 13 cm of the walls or each other. This distance, chosen as half the wingspan of *T. brasiliensis* [34], indicates a risk of collision.

The numerical results are visualized in Fig 2 and the median for each variable is listed in Table 1. Additional details are provided in Table A in S1 File.

Fig 3 shows the distance between agents and obstacles in the simulation for the default settings and the settings that ignore the echoes of the bodies of others and their calls. We excluded the first 50 simulation steps to give agents time to adjust to their random initial positions relative to the walls or other agents. In Fig 3, we also compare the data for the agent-based simulation with the data on interbat distances as reported by Weesner et al. [52] for *T. brasiliensis* emerging from a cave.

In the branched arena, the median number of wall collisions per simulation was 3. The median simulation duration was 164 steps, or approximately 6.6 seconds. Bats collided with the walls at a median rate of $0.72 \times 10^{-3}$ collisions per second per bat. The distribution of agent-to-wall distances was broad and difficult to summarize with a single value (Fig 3A). Throughout this results section, we use the mode of this distribution to compare distances across conditions, while detailed comparisons are best interpreted through the visualizations. In the branched arena, the median mode of agent-to-wall distances was about 1.6 m. Agents

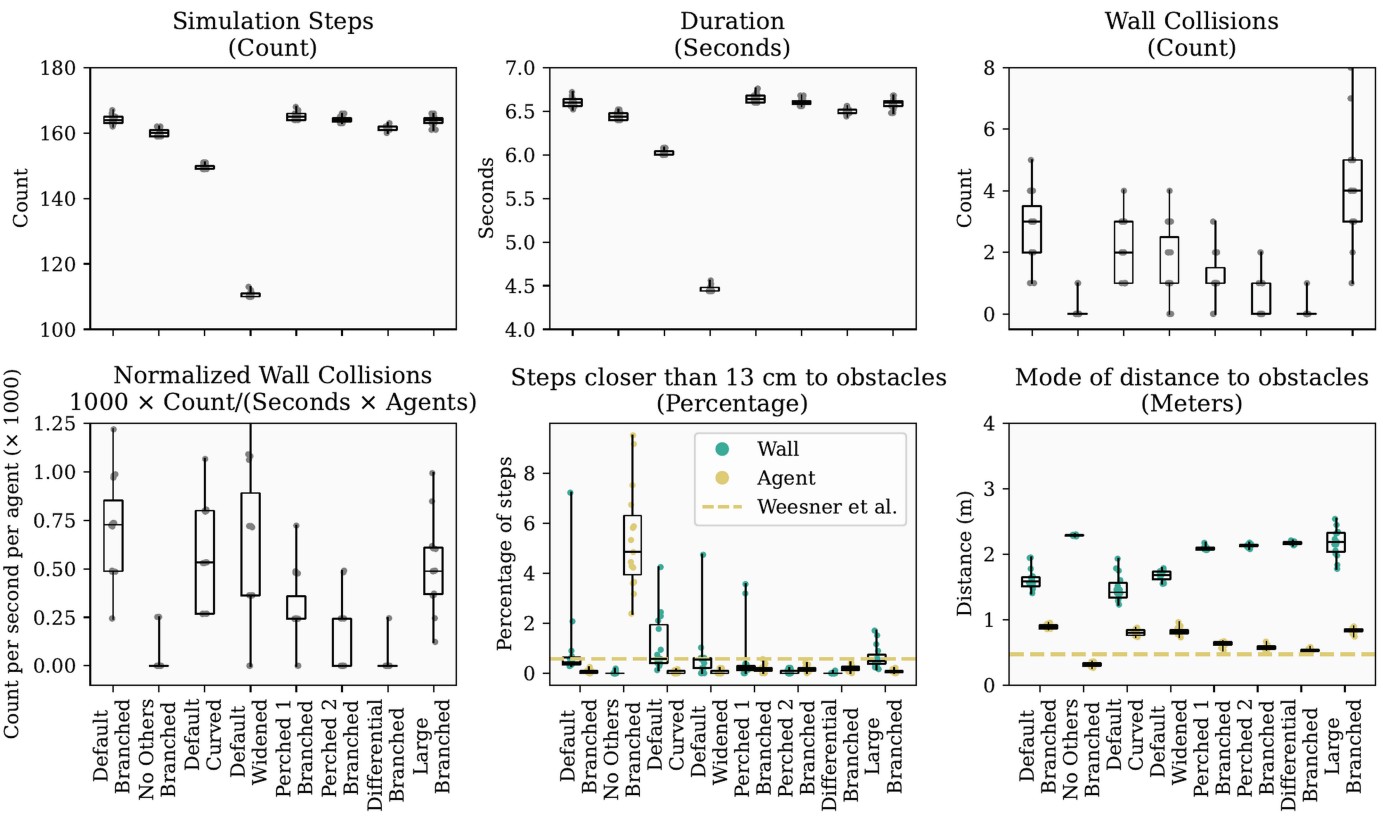

**Fig 2. Visualization of the results from the simulations in this paper across 15 simulation runs.** The labels on the x-axis indicate the simulation settings. *No Others* refers to simulations in which agents did not perceive other agents, *Perched 1* and *Perched 2* refer to simulations in which 25% and 50% of the wall reflectors were occupied by perched agents, *Differential* refers to a condition in which the agents could differentiate between echoes from walls and other acoustic inputs, and *Large* refers to a condition in which the arena was scaled up and the number of agents doubled to 50. Each label also indicates the arena used. The horizontal dashed line in the last two panels shows the data from Weesner et al. [52].

**Table 1. Summary of numerical results from the simulations in this paper.** All values reported are the median across 15 simulation runs. See Table A in S1 File for minima and maxima. The top line of the table header indicates the simulation settings: *No Others* refers to simulations in which agents did not perceive other agents, *Perched 1* and *Perched 2* refer to simulations in which 25% and 50% of the wall reflectors were occupied by perched agents, *Differential* refers to a condition in which the agents could differentiate between echoes from walls and other acoustic inputs, and *Large* refers to a condition in which the arena was scaled up and the number of agents doubled to 50. The bottom line of the table header indicates which arena was used. The table lists the mode (md) of the distribution for agent-to-agent and agent-to-wall distances. It also includes the duration of each simulation in simulated seconds and steps. The rows Agent < 13 cm and Wall < 13 cm indicate the percentage of time steps during which an agent was within 13 cm of another agent or a wall, respectively. Finally, the number of wall collisions is reported both as raw counts and as a normalized (N) value (collisions per agent per second).

| Statistic | Default Branched | No Others Branched | Default Curved | Default Widened | Perched 1 Branched | Perched 2 Branched | Differential Branched | Large Branched |
|---|---|---|---|---|---|---|---|---|
| Agent Dist. (md) | 0.901 | 0.311 | 0.803 | 0.822 | 0.641 | 0.580 | 0.525 | 0.843 |
| Wall Dist. (md) | 1.583 | 2.292 | 1.419 | 1.682 | 2.087 | 2.131 | 2.163 | 2.188 |
| Dur. | 6.600 | 6.440 | 6.040 | 4.440 | 6.640 | 6.600 | 6.480 | 6.600 |
| Steps | 164 | 160 | 150 | 110 | 165 | 164 | 161 | 164 |
| Agent < 13 cm | 0.036 | 4.850 | 0.000 | 0.000 | 0.177 | 0.143 | 0.183 | 0.055 |
| Wall < 13 cm | 0.431 | 0.000 | 0.578 | 0.545 | 0.212 | 0.000 | 0.000 | 0.494 |
| Wall Coll. | 3 | 0 | 2 | 1 | 1 | 0 | 0 | 4 |
| Wall Coll. (N), ×1000 | 0.727 | 0.000 | 0.533 | 0.364 | 0.242 | 0.000 | 0.000 | 0.488 |
| Agents | 25 | 25 | 25 | 25 | 25 | 25 | 25 | 50 |

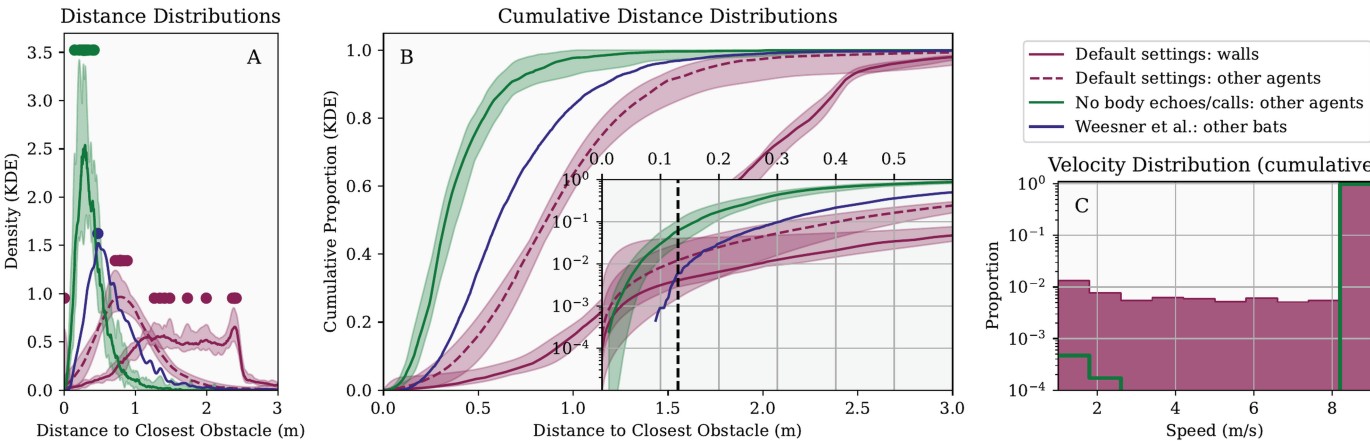

**Fig 3. (A) The purple line shows the distribution of agent-to-wall distances in the branched arena across 15 simulation runs.** The dashed purple line represents the distribution of inter-agent distances. The green line corresponds to a control condition in which agents did not respond to calls or echoes from other agents. As noted in the main text, this condition demonstrates that agents used calls and echoes from others to avoid collisions, resulting in a more dispersed distribution when these acoustic inputs were included in the model. The blue line represents empirical data from *T. brasiliensis* emerging from a cave, as reported by [52]. Dots on each line indicate the mode of the respective distributions across the 15 simulations. (B) The same data as in (A), but shown as cumulative distributions. The inset zooms in on the far left of the main panel to highlight differences at smaller obstacle distances, with the vertical line marking half the wingspan of *T. brasiliensis* (13 cm). (C) Distribution of agent velocities. In both panels (A) and (B), variability across simulation runs is depicted by shaded regions indicating the range (minimum and maximum) of each distribution across the 15 simulations, while solid lines represent the median distribution.

approached the walls closer than 13 cm (half the wingspan of *T. brasiliensis* [34]) in 0.43% of simulation steps (median across simulations).

The limited number of wall collisions implied that most of the time, the agents flew at speeds of 9 m/s. Indeed, in the current simulations, their speed was only reduced following a collision. Fig 1C shows the distribution of the speed for all simulated steps. This reveals that the probability of a given agent flying at 9 m/s at a given simulation step was almost 1.

We also analyzed inter-agent distances. The median mode of these distances was 0.90 m, meaning agents were typically closer to each other than to the walls. Additionally, agents rarely approached within 13 cm of one another, occurring in only 0.03% (proportion = 0.0003) of simulation steps (median across simulations).

These results were similar in the two other arenas tested—the curved and widened arenas. The median number of collisions per run was 2 in the curved arena and 1 in the widened arena. The collision rates, expressed as the number of collisions per bat per second, were $0.53 \times 10^{-3}$ and $0.36 \times 10^{-3}$, respectively. Agents approached walls to within 13 cm in 0.57% and 0.54% of simulation steps (median across simulations) in the curved and widened arenas, respectively. They spent 0.00% and 0.00% of steps within 13 cm of another agent (median across simulations).

To test whether agents in our simulation were actively repelled from each other by calls and echoes reflected from other bodies, we ran simulations excluding these acoustic inputs. Under these conditions, the median mode of inter-agent distance decreased to 0.31 meters, and agents were within 13 cm of each other in 4.8% of simulation steps (median across simulations) in the branched arena. These results confirm that agents were able to steer away from each other using calls and echoes, without explicitly identifying them as such.

Fig 4A shows how often each acoustic input was the loudest for each time step in the simulations. As is the case for the visualization in Fig 3, this graph is based on the strength of the acoustic input before converting it into a waveform. Fig 4A reveals that in most time steps,

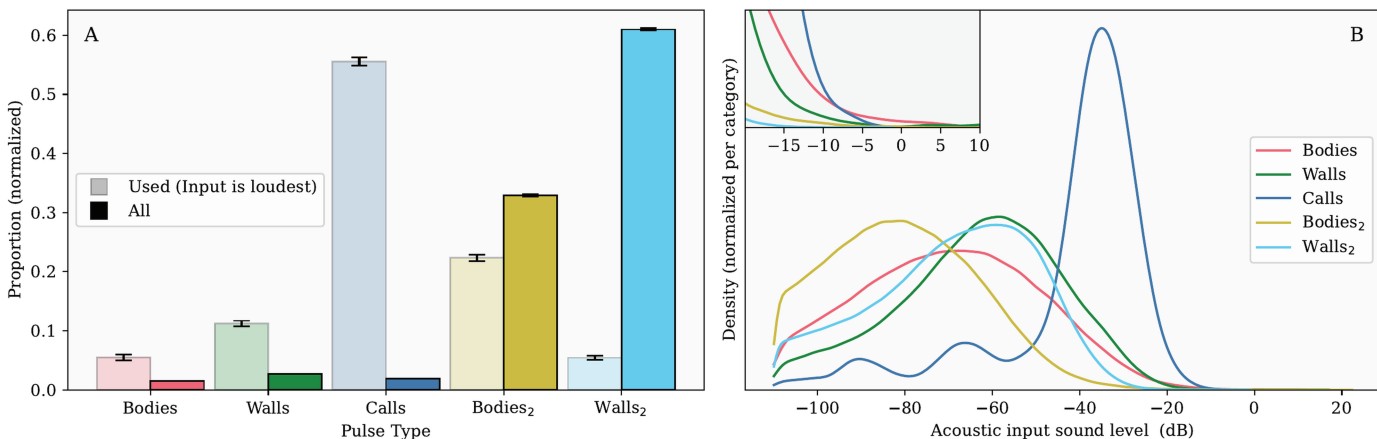

**Fig 4. Graphs showing the number of acoustic inputs calculated and used, as well as their relative sound levels.** (A) The distribution of loudest acoustic inputs (light bars) and all acoustic inputs (darker bars). Each set was normalized to sum to one. The error bars on each bar indicate the standard errors of the mean. (B) The distribution of the sound levels of each type of acoustic input for the default simulation settings. The inset shows details of the right side of the main plot.

the loudest acoustic input was either a call from other agents or an echo reflected off another agent's body of such a call. These acoustic inputs were most often used to update the agents' angular velocity. The self-generated body echoes were used the least.

Fig 4A also shows the number of acoustic inputs of each type calculated during the simulations (darker bars, normalized to sum to 1). This reveals that the simulated agent primarily received acoustic input in the form of echoes from the walls, particularly those generated by other agents. This is because the number of reflectors that form the walls far outnumbers the number of agents. In addition, each agent receives second-hand body echoes from multiple other individuals.

Fig 4B shows the distribution of the amplitudes of the different acoustic inputs. This graph shows that the calls made by others were the loudest on average. In contrast, on average, the secondary body echoes were the weakest input to the agents. This graph, together with the occurrence data plotted in Fig 4A, can explain the differences in the use of the different types of acoustic input by agents.

The agents seldom used indirect echoes from the walls despite these being the most prevalent echoes in the simulation. This is explained by the fact that these echoes were relatively weak compared to the calls. Similarly, direct echoes from others' bodies were seldom used. These echoes could be relatively loud (see inset Fig 4B). However, the agents received very few of these echoes. Moreover, likely spatial configurations of strong body echoes also led to loud, direct calls or indirect body echoes. This seems to be confirmed by the finding that both were used to update the rotation of the agents, despite these inputs being very scarce. Interestingly, while secondary body echoes were, on average, less loud than direct body echoes, their maximum values slightly exceeded those of the self-generated body echoes. This can be explained by the fact that these can be generated by another agent closer to the agent from which the echo returns. In turn, this led to these secondary body echoes being used quite often. Finally, the agents also used the echoes from the walls. This is explained by these inputs being somewhat prevalent and strong.

## Adding social calls

In the simulations presented so far, all bats were flying through the corridor. However, when *T. brasiliensis* emerges from a cave, not all individuals leave simultaneously. Therefore, while some bats are making their way to the cave's exit, others are still perched on the cave walls. Likewise, when bats reenter the roost, bats that arrive later will encounter a situation where some bats have already perched on the walls. Although we are unaware of recordings of *T. brasiliensis* roosting, in the lab, perched bats emit social calls [35]. Hence, it is possible that bats also emit calls while perched in caves or other roosts. In *T. brasiliensis*, social calls spectrotemporally resemble echolocation calls [7,35]. Therefore, calls emitted by perched bats could affect flying bats.

To assess how social (or echolocation) calls from perched bats affect the behavior of our agents, we ran simulations in which we assumed perched bats were scattered across the walls of the arenas. In particular, we randomly placed the perched agents in 25% of the positions of the reflectors that make up the walls. The arena consisted of 1968 reflectors covering an area of about 200 m$^2$. Placing agents at 25% of the reflector positions implies that we modeled about 2.5 agents per square meter. To our knowledge, there are few estimates of the densities of roosting *T. brasiliensis*. However, 2.5 agents per square meter is a very low density compared to available observations made on *T. brasiliensis*.Tuttle and colleagues [36] mention that 2000 bats per square meter are roughly the average adult roosting density in *Tadarida*. Carpenter et al. [37] estimated a density of about 400 bats per square meter for a fruit bat. Although these densities are substantially higher than the density modeled here, it should be noted that we only model bats producing a social call during a single time step (i.e., in a period of 40 ms). In addition, bat densities are likely to differ substantially between regions of a roost.

The orientation of these agents was randomized both in azimuth and in elevation. These agents were assumed to emit a single pulse during each simulation time step. The positions of the perched bats were randomized for each simulation step. Because social calls and echolocation calls resemble each other in *T. brasiliensis*, we modeled these social calls the same way as the echolocation calls were modeled (see Methods for details). The results of these simulations are depicted in Fig 5 and listed in Table 1 for the branched arena.

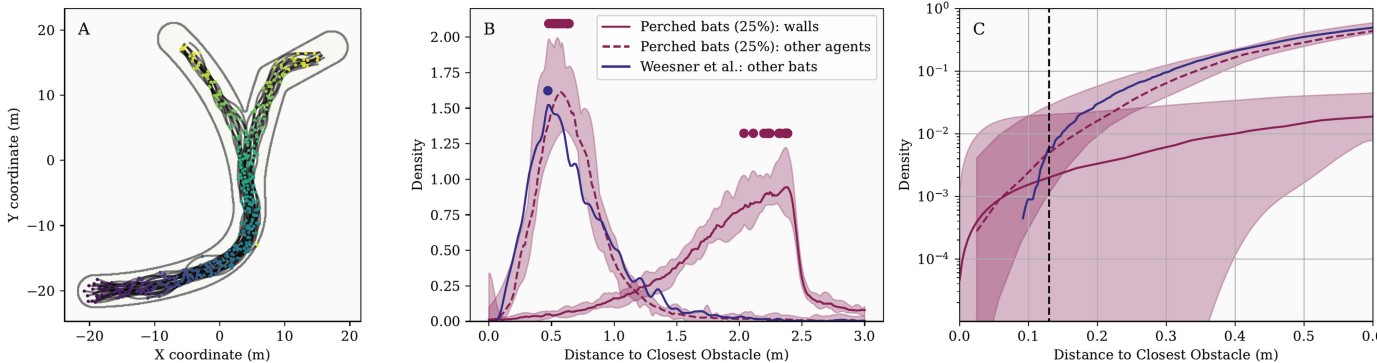

**Fig 5. Results for a variation of the simulation that included perched bats on the corridor walls.** (A) Example of the paths taken by the bats (1 run out of the 15 iterations). (B) Distributions of the distances between agents compared with the interbat distances observed by Weesner et al. [52]. (C) The same data as in for panel (B) but visualized as cumulative distributions and focused on small distances. Variation across simulation runs is shown in the same way as in Fig 3.

Placing perched agents on the corridor walls increased wall avoidance, leading to a more compact distribution of agents. The median inter-agent distance mode was 0.64 m, aligning more closely with the approximately 0.5 m observed in [32,52] than simulations without perched agents. However, agents did not entirely avoid the walls; the median number of wall collisions per simulation run was 1 (Table 1). Agents approached other agents and walls within 13 cm in only 0.17% and 0.21% of simulation steps, respectively. Thus, agents were more closely packed while maintaining greater wall distance compared to simulations without perched agents.

To assess whether the results were sensitive to the number of agents placed on the walls, we conducted a second set of 15 simulations, doubling the number of perched agents by placing them at 50% of the reflector positions. Example paths and distance distributions for this variation are presented in Fig E in S1 File. As shown in Table 1 and Fig 2, increasing the number of perched agents had little impact on agent spacing. However, it further reduced wall collisions: the median number of wall collisions was 0, and agents never came closer than 13 cm to the walls.

## Differentiating between inputs

In the simulations presented above, we intentionally kept the agents' processing of the acoustic input very simple. However, more involved processing of the acoustic field might increase agents' (and bats') ability to avoid the walls. For example, the avoidance of walls might be improved by allowing agents to process different classes of acoustic input separately. In the current model, agents do not differentiate between the acoustic inputs. However, the (directly picked up) calls emitted by others might differ sufficiently in their spectrotemporal properties from wall echoes (be they self-generated or caused by others' calls). Echoes from extended obstacles such as cave walls can be expected to be prolonged [38], in contrast to short and loud calls (see, for example, the echoes from extended vegetation targets reported by Yovel et al. [39]). If bats could distinguish calls from echoes, these two classes of acoustic input could be used differently to control flight, allowing for a more refined use of the acoustic field. Note that this does not require bats to isolate their self-generated echoes.

To assess this hypothesis, we ran simulations in which we equipped the agents with the ability to distinguish echoes from the walls from calls and echoes generated by the bodies of other agents. We implemented this by calculating the acoustic input for each agent twice: once including all acoustic inputs and once omitting the calls and echoes generated by the bodies of other agents. Next, we calculated the maximum value of the resulting cochlear output for these two variations of the acoustic input. Agents only responded to the complete acoustic input when the associated maximum value was substantially higher than for the cochlear output calculated on the acoustic input, omitting calls and echoes generated by the bodies of other agents. In other words, if the acoustic inputs that were not wall echoes were loud enough, the agents responded by avoiding them. Otherwise, the agents would react only to the wall echoes. Therefore, it should be noted that this implementation does not propose a signal-processing mechanism by which bats could isolate wall echoes from the other input.

In Fig 6, we show the results of simulations (15 runs) in which we have equipped the agents with the ability to distinguish echoes from the walls from calls and echoes generated by the bodies of other agents. The agents respond to the echoes from the walls as before. They respond only to calls and body echoes when these become loud enough. The median mode of the distance between bats was 0.52 m, close to the distance of about 0.5 m observed by Weesner et al. [52] and Theriault et al. [32]. The median number of collisions with the walls per simulation was zero (Table 1), and the agents did not spend any of their time nearer than

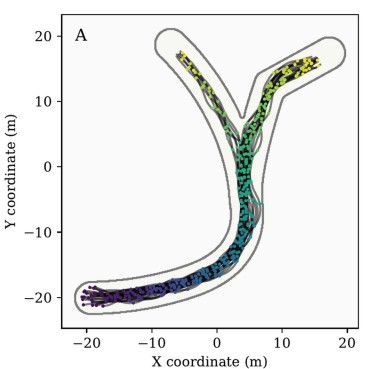
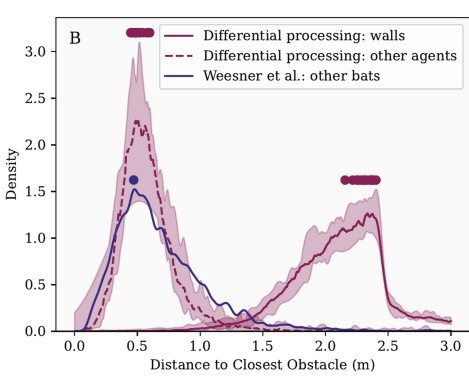
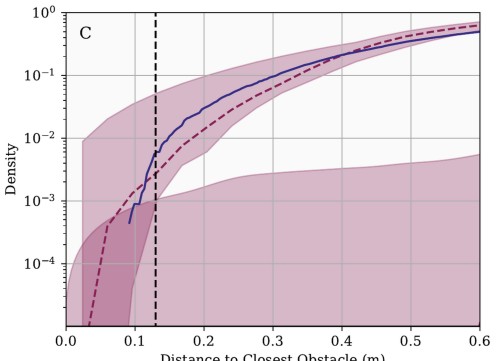

**Fig 6. Results of simulations in which the agents responded differently to wall echoes and other types of acoustic input.** Avoidance of the second class of acoustic inputs was triggered when these became too loud. (A) Example of the paths of the agents in a single run in the branched arena. (B) Distribution of closest distances between the agents in the simulations and the data of Weesner et al. [52]. (C) The same data as in panel (B) is visualized as a cumulative distribution showing distances below 0.6 m. Variation across simulation runs is shown in the same way as in Fig 3.

13 cm to the walls. Other agents were approached within a 13 cm distance in 0.18% of the simulation steps.

## Increasing the number of agents

In the simulations mentioned so far, we used 25 agents. As mentioned, this was done to reduce the computational load of the simulations, which increases non-linearly with the number of agents. However, to verify that the mechanism proposed in this paper scales to more agents, we ran selected simulations with an increased number of agents. We scaled the branched arena by a factor $\sqrt{2}$ in the horizontal (x and y) dimensions and doubled the number of agents to 50 to keep the density of agents constant. Fig 7 shows the results of the simulations with 50 agents in the scaled-up branched arena.

In these simulations, we registered a median of 4 wall collisions per run. However, since the number of agents was doubled, this implied a collision rate of $0.485 \times 10^{-3}$ collisions per bat per second, i.e., an almost identical collision rate than found for 25 agents in the same (scaled) arena (Table 1). Agents approached each other slightly more closely than when simulating 25 agents. However, the steps in which the agents approached each other to within 13 cm decreased from 0.090% to 0.055%. The amount of time spent closer than 13 cm to a wall increased slightly from 0.30% to 0.49%. Hence, increasing the number of agents did not increase the collisions with the walls and actually decreased the amount of time agents spent closer than 13 cm to each other. Hence, doubling the number of agents had no great impact on the performance.

## Discussion

Jamming has been investigated primarily in the context of bats foraging within the same airspace. When hunting, bats presumably must distinguish the self-generated prey echoes from the echoes and calls generated by nearby conspecifics. This is required to extract positional information about the target, which is likely critical for effective foraging, e.g., [3–5].

Several studies (but not all) found that foraging bats change the frequency content of their signals in the presence of conspecifics (See [16] for references). This has been interpreted as an attempt to distinguish their echoes from those of others. However, using a computational model Mazar et al. [16] suggested that bats hunting near each other might experience less

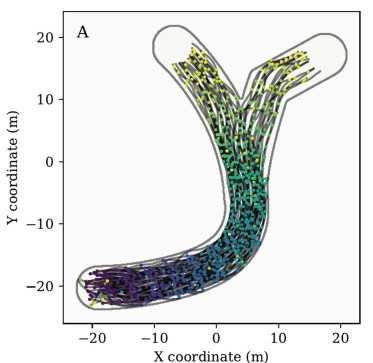
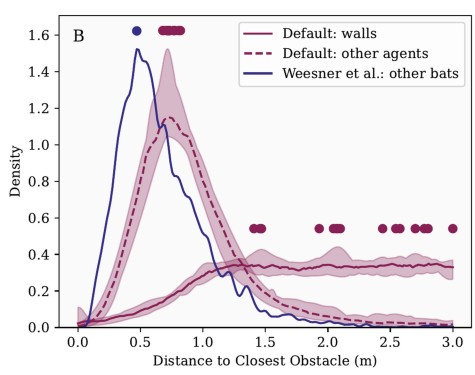
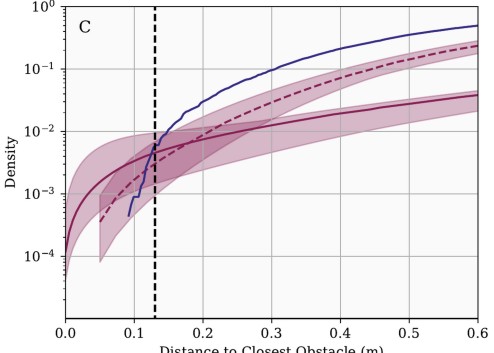

**Fig 7. Results of simulations in which the number of agents was doubled, and the arena was scaled.** (A) Example of the paths of the agents in a single run in the branched arena. (B) Distribution of closest distances between the agents in the simulations and the data of Weesner et al. [52]. (C) The same data as in panel (B) is visualized as a cumulative distribution showing distances below 0.6 m. Variation across simulation runs is shown in the same way as in Fig 3.

jamming than previously thought. They suggested that intrinsic changes in call design and echo loudness as the bat approaches the target are sufficient to unmask echoes and avoid confusion between self-generated prey echoes and other signals. As a result, actively changing the call's spectral content was considered ineffective. Alternative explanations for some of the jamming avoidance responses observed in bats have been suggested [4,16].

This paper presents a computational model of another situation in which extensive jamming is thought to occur. We modeled bats leaving (or entering) a roost. *T. brasiliensis* is the most iconic bat species that leaves [9] and enters [13] roosts, often in dense and large groups. However, other species have also been documented to emerge in groups from roosts, such as caves [30,40]. In this scenario, the inherent mechanisms that prevent jamming during hunting, as proposed by Mazar et al. [16], would be ineffective. First, their simulated density of bats was lower than that observed in bats emerging from caves: they simulated up to 20 bats per 100 m² (their simulations were performed in 2 dimensions). In contrast, Lin et al. [30] reported a density of 0.4 bats/m³, and we calculated that Weesner et al. [52] observed an average density of about 2.2 bats per cubic meter. Second, the variation in calls used by bats in a swarm appears to be lower than that used by foraging bats. Gillam et al. [7] found that *T. brasiliensis* mainly used two calls with similar frequency ranges and durations (but different spectrotemporal details) during emergence. In contrast, during the search for prey, *T. brasiliensis*'s call varied in duration from about 7 to less than 1 ms, while the spectral content varied from near constant frequency to broadband [41]. Hence, *T. brasiliensis* does not appear to use the full range of possible changes in the spectrotemporal makeup of its calls to cope with jamming during emergence.

Beleyur et al. [42] previously presented simulations of bats flying in groups. They suggested that swarming bats might be able to detect their self-generated echoes. In their study, an echo was considered detectable if its envelope exceeded that of any masking echoes, at least for part of its duration. However, they did not demonstrate how bats could *distinguish* their echoes from those generated by conspecifics. Indeed, these authors suggest (but do not test) that individual differences in the spectral content of calls could be used for this. They also did not demonstrate that such detection alone supports obstacle avoidance.

More recently, Mazar et al. [38] described a 2D simulation study that is closely related to the current one. These authors simulated bats (modeled after *Pipistrellus kuhli* and *Rhinopoma microphyllum*) finding the exit to a cave while relying solely on echolocation. In contrast to

our study, they assumed that bats could distinguish between echoes resulting from their own calls and those generated by others' calls. Additionally, their model assumed that bats could estimate the distance and horizontal angle of origin of these echoes. This spatial information was used for path planning and dynamically adjusting pulse rates and flight speeds. Furthermore, their model assumed that calls from other bats only mask echoes but cannot be mistaken for them.

We suggest that the assumptions underlying our simulations are less stringent. Our approach does not assume that bats can isolate self-generated echoes or derive spatial information about obstacles and conspecifics. Our agents do not estimate the relative position of wall reflectors or other agents and treat calls emitted by others like echoes. Mazar et al. [38] did explore a condition in which the assumption of self-echo isolation was dropped. However, even under this condition, they retained the assumption that echoes could be localized and that calls from others only act as masking signals. It is necessary to be cautious in directly comparing results between the study of Mazar et al. [38] and the current simulations. Among other settings that differed, Mazar et al. [38] used different arenas for their bats. With this caveat, we include a brief comparison here. When simulating 20 bats, they recorded about 0.1 collisions per bat per second (with the walls) under the best performance conditions. In our simulations, we registered between 0 and $0.5 \times 10^{-3}$ collisions per bat per second (Table 1). Hence, performance in the current simulations did not seem worse, using fewer assumptions. Consequently, we argue that the current simulations demonstrate that the assumptions regarding the acoustic discriminations and inferences bats need to make when emerging in groups from caves or roosts can be further relaxed beyond those proposed by Mazar et al. [38].

Performance in the current simulations was not perfect. We observed wall collisions and a small proportion of simulated steps where agents came within half a wingspan of the walls or each other (Table 1). On the one hand, it is likely that bats, if relying on the collective sound field without isolating their self-generated echoes or distinguishing between acoustic inputs, use more sophisticated acoustic processing than that modeled here. Additionally, bats may utilize other sources of information not included in these simulations, such as spatial memory, which could enhance wall avoidance—and airflow receptors on their wing membranes [43], which may help detect the proximity of other bats. On the other hand, reanalyzing the data by Weesner et al. [52], we found that *T. brasiliensis* was observed to fly closer than half a wingspan (13 cm, as used in this paper) to each other in 0.55% of the observed interbat distances. This suggests that the half-wingspan criterion may be more conservative than the distances real bats maintain. Moreover, to our knowledge, it is unknown whether and how often *T. brasiliensis* (or other bats negotiating caves in groups) collide with walls.

The simulations described in this paper suggest that the soundscape collectively produced by bats in a swarm provides sufficient information about the spatial distribution of others and the walls without the need to isolate self-generated echoes. The agents relied solely on the interaural level difference of the loudest part of the acoustic input. This suggests that the jamming problem experienced by bat groups flying in confined spaces may not need to be solved for successful sensorimotor behavior and obstacle avoidance. As such, our results could be interpreted as expanding the modeling results of Mazar et al. [16,38] by further reducing the challenge jamming poses to swarming bats and reducing the requirements posed for successful swarming. Moreover, the current paper also expands on this previous work by shifting the perspective on jamming itself. Rather than framing overlapping signals solely as a source of interference, as done in the simulations presented in [38,44], our findings highlight that these signals can also carry useful information, reframing the paradigm from one of conflict to one of cooperative signal processing.

In our current simulations, the initial density of the group was modeled based on observations by Weesner et al. [52] (Fig 8). However, the agents in our simulations ended up spacing out further (Figs 3 and 8, median of mode of distance to other agents: 0.52 to 0.90) than the bats in the swarm filmed by Weesner et al. [52] (mode of distance distribution: 0.47). This might be explained by the fact that, in our simulations, agents are only equipped with a mechanism to avoid the walls and each other. There is no mechanism that keeps the agents together (apart from the repulsion from the walls). In simulations where wall repulsion was increased by either adding perched agents or allowing agents to distinguish between classes of acoustic input, the spacing of the agents was better matched to those of the bats (Figs 5 and 6 and Fig E in S1 File).

We could not find an acoustic mechanism that keeps the simulated agents together without repulsing walls. Observations of *T. brasiliensis* show that these bats must have a mechanism to maintain cohesion outside of the roost. Even when they have emerged from a cave, *T. brasiliensis* remain together in dense groups that form the serpentine columns for which they are well known [7,10]. However, so far, this group cohesion has only been observed in emerging bats outside the roost. To our knowledge, there are no data on the density and behavior of bat swarms in roosts, except for the data on *M. fuliginosus* presented by Lin et al. [30], who found much lower densities than Weesner et al. [52] for *T. brasiliensis*.

Hence, while we compare the behavior of our agents with the only data available, it should be noted that the data and the simulations pertain to two different situations: inside and outside the roost. Outside the roost, group cohesion is likely to be exhibited to avoid predation. Brighton et al. [45] described how bats within a tight column are less susceptible to being attacked by falcons, which attack lone bats disproportionately. These authors also speculate that the reduced risk of predation may explain why the swarm loses its coherence away from the cave. If diurnal predators close to the roost are the reason why *T. brasiliensis* maintains tight group cohesion, bats emerging after sunset might show less cohesion. In accordance with this prediction, Wilkins [10] discussed how bats that emerge after sunset do so in diffuse groups rather than in serpentine columns. Therefore, light levels are probably important to determine whether emergence occurs in a diffuse pattern or in tight groups. This opens up the possibility that the tight groups of *T. brasiliensis* emerging before sunset are maintained using vision instead of echolocation, akin to the mechanisms underlying murmurations in birds, e.g., [46,47].

In sum, while our agent tends to spread more widely than the bats observed by Weesner et al. [52] (in the absence of additional repulsion by the walls), it is possible that bats'

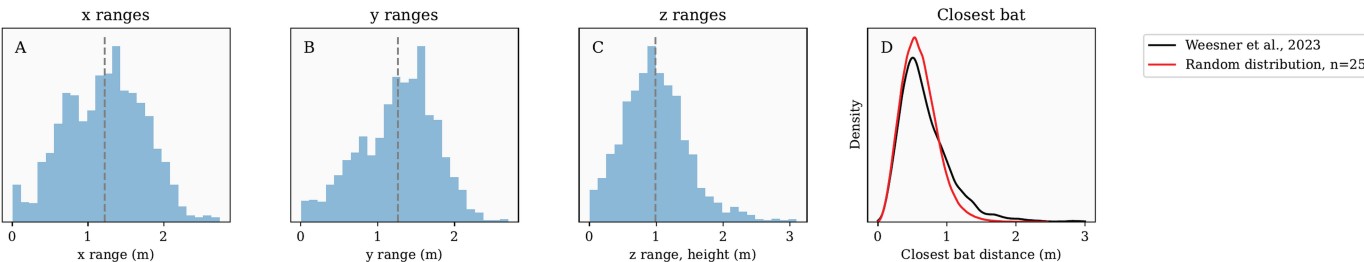

**Fig 8. (A-C) The range in the x, y (horizontal directions) and z (vertical direction) direction of the bats in the video data of *T. brasiliensis* emerging from a cave collected by Weesner et al.** [52]. The vertical lines represent the mean of each distribution. (D) Distribution of the distances of each bat to its closest neighbor in the video data of Weesner et al. [52]. Overlaid is the simulated distribution of distances between 25 agents placed in a cylindrical volume of about 20 m³ (i.e., a cylinder with a diameter of 5 m and height of 1 m).

density outside the roost does not match the outside behavior observed by Weesner et al. [52]. Moreover, the tight serpentine columns appear to occur only under conditions in which vision is available to maintain group cohesion.

The current paper suggests the feasibility of obstacle avoidance in a swarm without the need to detect and identify self-generated echoes. As such, this paper provides an alternative hypothesis about how swarming bats within roosts deal with the cocktail party nightmare. However, the simulations in this paper cannot establish whether this approach is actually employed by bats or if it is the precise mechanism used. If it is confirmed that bat swarms exploit a mechanism similar to the one described here, this would constitute a different, new form of echolocation that does not rely so much on processing self-generated echoes, but rather on interpreting a collectively built soundscape.

## Methods

### Arena

We created different arenas to test the proposed strategy's ability to lead the agent swarm through the corridor, avoiding obstacles. The arena walls consisted of points 0.1 meters apart in the $x$ and $y$ directions. The vertical position of each point $z$ was uniformly distributed between -1 and 1 meters. The corridors were about 5 m wide.

The arenas contained a cylindrical starting region with a volume of about 12.5 $m^3$ (diameter: 4 m, height: 1 m). At the beginning of each simulation run, each agent appeared in a random location $(x, y, z)$ within the starting region with a jittered heading. The vertical position $z$ of the agent was randomly selected between -0.5 and 0.5 meters. This height variation was selected based on a reanalysis of the data of Weesner et al. [52]. These authors used a thermal camera to film the emergence of *T. brasiliensis*. They extracted the position of each bat in each frame with respect to the camera. In Fig 8A, we plot the distribution of flight height ranges (difference between the lowest and highest bats) within a single frame. We found that, on average, the vertical spread of bats tracked by Weesner et al. [52] was approximately 1 meter. In addition to a starting region, the arena also contained one or two cylindrical exit regions. Agents entering this region were removed from the simulation to model bats leaving the arena or cave.

### Agent density

To simulate bat groups with realistic densities, we reanalyzed the data provided by Weesner et al. [52]. As mentioned, these authors recorded the emergence of *T. brasiliensis* and extracted the position of each bat in each frame with respect to the camera. Fig 8A-C show the distribution in the range of bats in the x, y, and z (height) directions for each frame. This figure shows that, on average, the bats filmed by Weesner et al. [52] occupied a volume of $1.21 \times 1.26 \times 0.99$ m. We also extracted the distance to the nearest conspecific for each bat in each frame. This distribution is shown in 8D. The average distance between a bat and its closest conspecific in the data reported by Weesner et al. [52] was 0.69 meters.

In our simulations, agents started in a cylindrical region of 12.5 $m^3$ (diameter: 4 m, height: 1). However, the width of the corridor was 5 meters. To ensure that the agent density, at least at the beginning of the simulations, was similar to the density observed by Weesner et al. [52], we simulated an increasing number of agents randomly placed in a cylindrical volume of about 20 $m^3$ (diameter: 5 m, height: 1m). For each number of agents assessed, we simulated 50000 iterations of randomly placed agents. For each iteration, we calculated the distance to the nearest other agent for each agent. We found that by placing 25 agents in a cylindrical

volume of approximately 20 $m^3$, the distribution of the closest distances between the agents matched well the empirical distribution derived from the data of Weesner et al. [52] (Fig 8D). Based on this analysis, we used populations of 25 agents in each simulation.

## Simulation of acoustic input

For each focal agent $i$ and each time step of the simulation, we calculated the strength (sound level in decibels) of the following acoustic inputs:

1. **Others' calls** The sound level at which agent $i$ perceived each of the calls emitted by each other agent $j$. The paper refers to these as *calls*.
2. **Body echoes** The sound level of the echoes generated by the emission of bat $i$ reflecting from the bodies of each other bat $j$. Throughout the paper, we refer to these as *body echoes*.
3. **Arena echoes** The sound level of the echoes generated by the emission of agent $i$ reflecting from each of the points $k$ that make up the arena. Throughout the paper, we refer to these as *wall echoes*
4. **Body echoes caused by the emissions of other agents** The sound level at which agent $i$ perceives the echoes of each other agent $j$'s body caused by the emissions of each agent $m$. Throughout the paper, we refer to these as *secondary body echoes* or *Bodies*$_2$ in figures
5. **Arena echoes caused by the emissions of other agents** The sound level at which agent $i$ perceives the echoes from each point $k$, making up the arena walls, caused by the emissions of each agent $m$. Throughout the paper, we refer to these as *secondary wall echoes* or *Walls*$_2$ in figures.

In the following sections, we describe in detail how the sound level of these acoustic inputs was calculated.

**Simulation of calls.** As mentioned, *calls* refer to the acoustic input received by a focal agent $i$ due to the emission of other agents $j$. We used the following equation to calculate its sound level,

$$C_{i,j} = E_{\theta(i,j)} + H_{\theta(j,i)} + 20 \times log_{10} \frac{1}{D(i,j)} + \alpha \times D(i,j) \tag{1}$$

In Eq 1, $E_{\theta(i,j)}$ refers to the emission directionality for azimuth and elevation direction $\theta$ of agent $j$ with respect to agent $i$. In other words, this term denotes the level of emission of the agent $j$ in the direction of the agent $i$. Conversely, the term $H_{\theta(j,i)}$ denotes the hearing sensitivity of agent $i$ in the direction of agent agent $j$ with respect to agent $i$. The next term in Eq 1 models spherical spreading as the call travels from agent $j$ to agent $i$ over a distance of $D(i,j)$, i.e., the distance of agent $i$ with respect to agent $j$. The last term models the atmospheric attenuation $\alpha$ at 40 kHz, i.e., -1.3 dB/m, Based on ISO 9613-1 [48,49]. This value was selected because Kloepper et al. [24] reported that *T. brasiliensis* uses calls sweeping down from approximately 50 to 30 kHz during roost reentry, and 40 kHz is halfway in this range. For each agent $i$, we applied the Eq 1 twice: once for the left ear and once for the right ear, with $H$ the hearing directionality for the left or right ear, respectively.

**Simulation of body echoes.** In this paper, body echoes refer to echoes received by a focal agent $i$ due to the bodies of other agents $j$ reflecting its own calls. We used the following equation to calculate the sound level of the acoustic input at each simulation step.

$$B_{i,j} = E_{\theta(j,i)} + H_{\theta(j,i)} + 40 \times log_{10}\frac{1}{D(i,j)} + \alpha \times 2 \times D(i,j) - T_b \qquad (2)$$

The terms in Eq 2 are similar to those in Eq 1. However, we use $E_{\theta(j,i)}$ to model the loudness of the emission of the focal agent $i$ in the direction $\theta$ of agent $j$ with respect to agent $i$. Furthermore, since the sound has to travel from agent $i$ to $j$ and back, we double the spherical spreading and atmospheric attenuation effect. Finally, we applied a random target strength $T_b$ between -10 and -44 dB for the bodies of the agents, based on the measurements reported in [42]. For each agent $i$, we applied Eq 2 twice: once for the left ear and once for the right ear, with $H$ the hearing directionality for the left or right ear, respectively.

**Simulation of wall echoes.** The wall echoes received by the focal agent $i$ are the result of its emissions reflecting from the points $k$ that make up the environment. The sound level of these echoes was calculated in a manner very similar to the calculation of the body echoes:

$$W_{i,j} = E_{\theta(k,i)} + H_{\theta(k,i)} + 20 \times log_{10}\frac{1}{D(k,i)} + \mathcal{S} \times log_{10}\frac{1}{D(k,i)} + \alpha \times 2 \times D(k,j) - T_w \qquad (3)$$

The target strength $T_w$ of the points making up the arena was set randomly to a value between -9 to -15 (i.e., centered around -12 dB with a range of $\pm 3$ dB). The center value was selected because it is somewhat lower than the target strength obtained for a concrete wall [49]. This random fluctuation in target strength reflects the assumption that different patches of the wall in a natural cave will reflect different amounts of sound energy to the agent, for example, due to their orientation. By selecting this range, we assumed that the rocky walls of a cave reflect more strongly than the bodies of bats. Again, this equation was used twice to simulate the wall echoes received by the left and right ears of the focal agent $i$. This was done using the directionality of the left and right hearing $H$, respectively.

The parameter $\mathcal{S}$ models the geometrical spreading of the echo toward the receiving bat. This parameter was set randomly for each echo in the range of 0 to 20, again based on the results reported in [49]. This implies that the reflection of the return echo can vary from planar (extended flat surface) to spherical (point). In contrast, we assume that an agent's call spreads spherically on the outward journey,

**Simulation of secondary body echoes.** We simulated the sound level of secondary body echoes, i.e., the echoes received by a focal agent $i$ from the bodies of agents $j$ caused by the emissions of agents $m$.

$$BB_{i,j} = E_{\theta(m,j)} + H_{\theta(j,i)} + 20 \times log_{10}\frac{1}{D(m,j)} + 20 \times log_{10}\frac{1}{D(i,j)}$$
$$+ \alpha \times D(m,j) + \alpha \times D(i,j) - T_b \qquad (4)$$

Eq 4 models sound propagation along two paths. First, it models the propagation from an emitting agent $m$ to the location of the agent $j$. Second, it models the resulting echo propagating from the position of the agent $j$ to the focus agent $i$. As we did for direct body echoes (Eq 2), we assumed a target strength $T_b$ of the agent bodies between -10 and -44 dB [42]. Like for the other components of the acoustic input to the focal agent $i$, we ran the equation twice: once for the left ear and once for the right ear.

**Simulation of the secondary wall echoes.** Finally, we modeled secondary wall echoes: the echoes perceived by the focal agent $i$ due to the calls of another agent $j$ reflecting from an arena point $k$. The equation used to model this was similar to Eq 4. In this equation, the parameter $\mathcal{S}$ was varied from 0 to 20, again to model variations of the geometrical spreading of the echoes.

$$WW_{i,j} = E_{\theta(k,j)} + H_{\theta(k,i)} + 20 \times log_{10}\frac{1}{D(k,j)} + \mathcal{S} \times log_{10}\frac{1}{D(i,k)} + \alpha \times D(k,j)$$
$$+ \alpha \times D(i,k) - T_w \qquad (5)$$

## Processing the acoustic input

The simulated loudness of the acoustic inputs (calls, body echoes, wall echoes, secondary body echoes, and secondary wall echoes) in the left and right ears was converted to waveforms using the following approach. We converted the simulated loudness to an impulse response representation. For the body and wall echoes, the time shifts for this representation were determined using the travel distance. No absolute timing was available in the simulation for the other acoustic inputs. The timing of other agents' calls determines when an agent receives these acoustic inputs. We assign each of these a random delay ranging from 0 to 25 ms to model this timing uncertainty. This reflects the assumption that an agent will receive these inputs at each interpulse interval, but the time they are received is random (with respect to the time at which other inputs are received), as groups of bats do not coordinate their calls [50]. The result of this operation is an impulse response representation of the acoustic input received by a randomly selected agent during a given step of the simulation. Fig 9A shows an example of this representation.

The impulse response representation was then converted to a waveform representation of the acoustic input by convolving it with an artificially generated batlike call sweeping down from 50 to 30 kHz over 9 ms. This resembles the calls used by *T. brasiliensis* while emerging from roosts [24]. The resulting waveform was run through a matched filter to align the

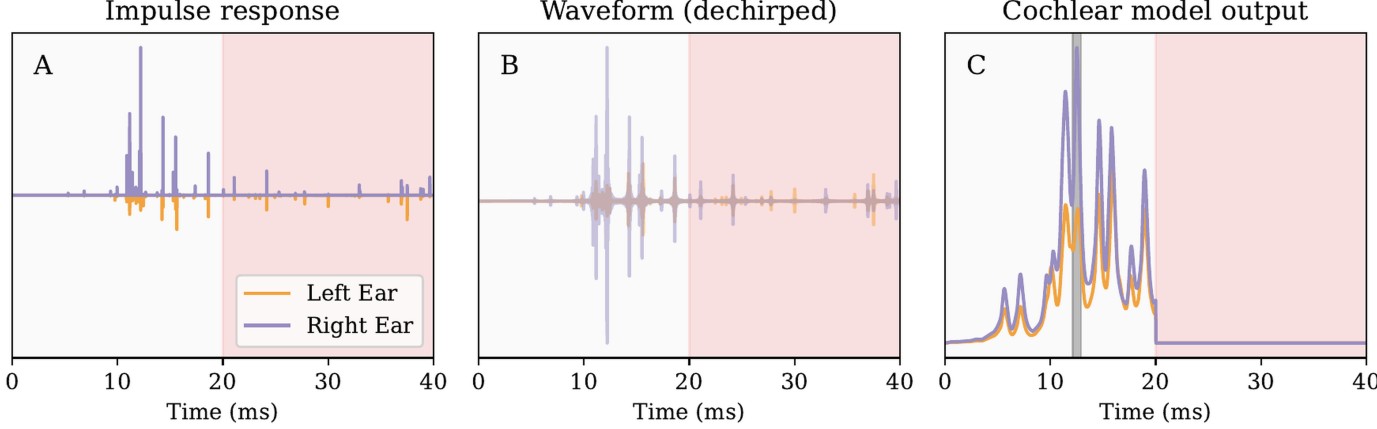

**Fig 9. (A) An example of the impulse response derived from the acoustic inputs simulated for a single bat at a specific step of the simulation (one ear).** The region shaded in red corresponds to echoes and calls computed but not used by the agents (or analyzed in the results). As explained in the main text, this was done to model a period in which the agents could process the acoustic input. (B) The waveform is obtained by convolving the impulse response in panel (A) with a batlike call. Again, the red region corresponds to acoustic input that was not processed. (C) The result of running the waveform in panel (B) through a cochlear model [27]. The gray shaded region indicates the parts of the signals integrated to obtain the echo strength in the left and right ears. The cochlear output in the second half of the interpulse interval was set to zero.

different frequencies in time. This operation is similar to dechirping the spectrogram or cochleogram [27]. An example is given in Fig 9B. Finally, the waveform was processed using the model presented by Wiegrebe [27] to obtain a simulated cochlear response. The output of the cochlear model was averaged across the frequency channels. An example is shown in Fig 9C.

Next, we determine when the cochlear output reaches a maximum in each ear. The first time was taken across the two ears. We integrated the cochlear output in the left and right ears in a 1-ms window around this time (see Fig 9C). The integrated values in the left and right ears were taken as the magnitude of the echo in the left and right ears. Finally, we calculate the interaural level difference by subtracting the value of the left ear from the right ear.

## Overview of simulation

Having discussed parts of the simulation, we now provide an overview of the flow of the simulation. Throughout this discussion, we refer to Fig 10, which depicts the flow of the simulation.

The main loop of each simulation run is depicted in Fig 10A-F. At the start of the simulation, the positions and orientations of the agents are initialized within the starting region of the arena (Fig 10A). Next, the order of the agents is randomized (Fig 10B). For each agent (Fig 10C), the controller is used to calculate its position and orientation update. Once all agents have been processed, the process is repeated in a different random order of agents. If more

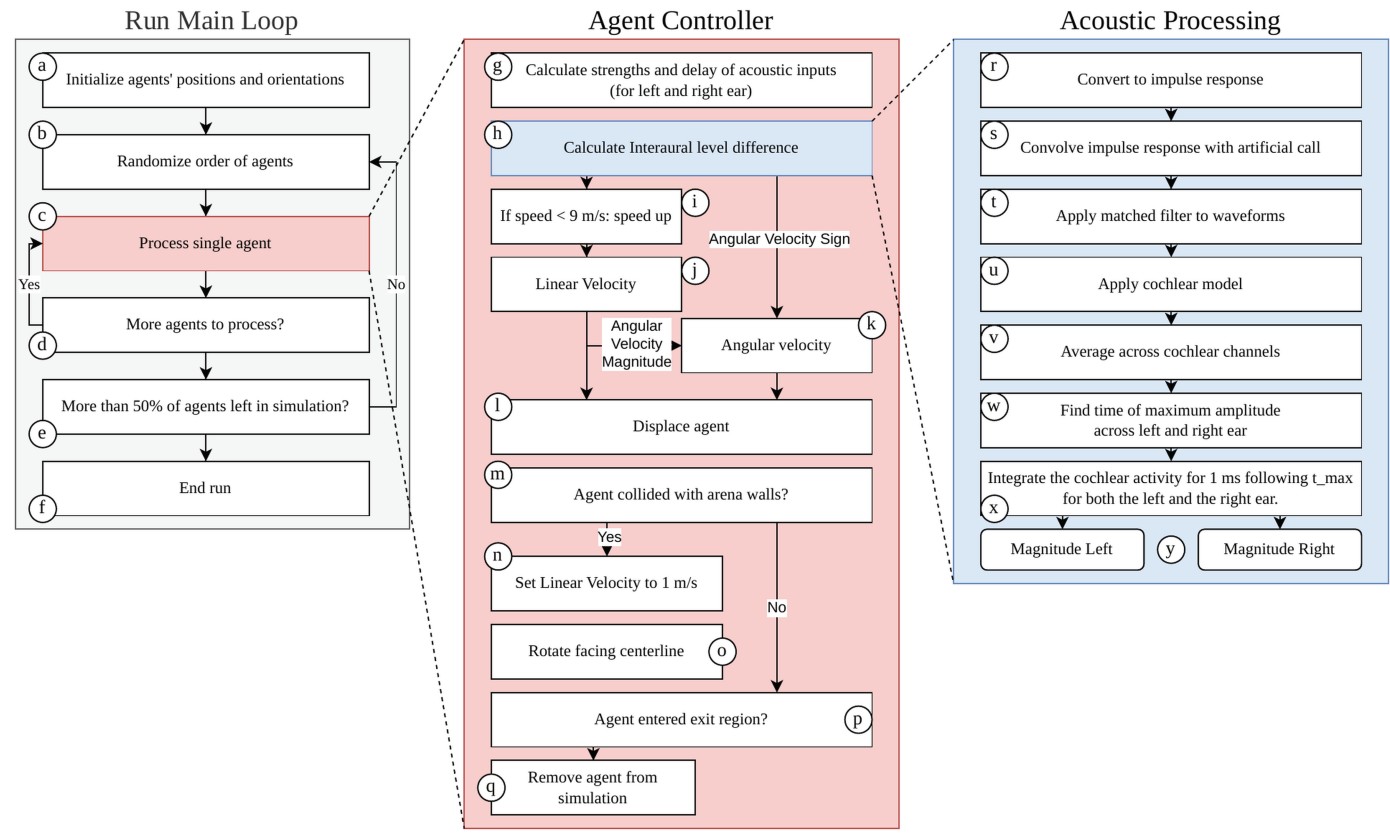

**Fig 10. Flowchart of the simulations.** See text for details.

than 50% of agents have left the arena by entering the exit region (Fig 10E), the simulation run ends (Fig 10F).

The agents' controller is shown in (Fig 10G-Q). The controller starts (Fig 10G) by calculating the strengths of the acoustic inputs (call, body echoes, wall echoes, secondary body echoes, and secondary wall echoes) using Eqs 1, 2, 3, 4, and 5. These are then processed (Fig 10H) using a method that will be described in more detail in the following. If the agent's current linear velocity is less than 9 m/s, its velocity increases (Fig 10I). The output of acoustic processing is a value for the interaural difference level. The sign of the interaural level difference is used to set the sign of the agent's angular velocity. The magnitude of the angular velocity is set on the basis of the agents' current linear velocity (Fig 10I-K). Using the current angular and linear velocity of the agent, the agent is displaced (Fig 10L).

Next, the simulation checks whether the agent has collided with the walls of the arena (Fig 10M). This is detected by assessing whether the agent has crossed the arena's boundaries. If the agent collided with the walls, its linear velocity is set to 1 m/s and rotated so that it faces the center line of the arena (Fig 10N-O). If the agent has reached the exit region, it is removed from the arena (Fig 10Q).

Finally, we explain the details of the acoustic processing (Fig 10R-Y). Acoustic processing starts with converting the loudness values of each echo received in the left and right ear into an impulse response representation (Fig 10R) (at a sample rate of 250 kHz). These representations are converted to waveforms by convolving them with a 9 ms long artificial bat call sweeping down in frequency from 50 to 30 kHz (Fig 10S). This frequency range and duration correspond to the call parameters Kloeper et al. [24] observed in *T. brasiliensis* during roost reentry. Next, we ran the waveforms through a matched filter to align the different frequencies of the call (and the echoes) in time (Fig 10T). This was done to be able to average across frequency bands in the next step. We processed the resulting waveforms using the cochlear model proposed by Wiegrebe [27] (Fig 10U). We used a model with 10 Gammatone filters as implemented in [51] centered around 40 kHz. The output of the cochlear model was then averaged across the frequency channels (Fig 10V). We looked for the time at which the averaged cochlear output reached a maximum in each of the two ears (Fig 10W). That is, we found the time when the overall maximum occurred. For both ears, we integrated the averaged cochlear output in a 1-ms window around the time the maximum occurred (Fig 10X). This resulted in a value for the magnitude in the left and right ear (Fig 10Y). The difference between these values was the interaural level difference used by the controller to set the direction of the agent's turn (Fig 10H).

## Supporting Information

**S1 File.** Supplementary file containing the items listed below, i.e., Table A and Figs A-G. (PDF)

## Author contributions

**Conceptualization:** Dieter Vanderelst, Herbert Peremans.

**Data curation:** Dieter Vanderelst.

**Formal analysis:** Dieter Vanderelst.

**Funding acquisition:** Dieter Vanderelst.

**Investigation:** Dieter Vanderelst, Herbert Peremans.

**Methodology:** Dieter Vanderelst, Herbert Peremans.

**Project administration:** Dieter Vanderelst.

**Resources:** Dieter Vanderelst.

**Software:** Dieter Vanderelst.

**Validation:** Dieter Vanderelst, Herbert Peremans.

**Visualization:** Dieter Vanderelst.

**Writing – original draft:** Dieter Vanderelst, Herbert Peremans.

**Writing – review & editing:** Dieter Vanderelst, Herbert Peremans.

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
