## [Decision Letter · Decision Letter 0]

4 Dec 2024

PCOMPBIOL-D-24-01732

How swarming bats can use the collective soundscape for obstacle avoidance

PLOS Computational Biology

Dear Dr. Vanderelst,

Thank you for submitting your manuscript to PLOS Computational Biology. After careful consideration, we feel that it has merit but does not fully meet PLOS Computational Biology's publication criteria as it currently stands. Therefore, we invite you to submit a revised version of the manuscript that addresses the points raised during the review process.

Both reviewers request some clarifications on details of the simulation modelling, as well as some of the motivation. I would suggest in particular that comments on simulation run numbers, and code quality, are addressed in the resubmission.

Please submit your revised manuscript within 30 days Feb 03 2025 11:59PM. If you will need more time than this to complete your revisions, please reply to this message or contact the journal office at ploscompbiol@plos.org. Please include the following items when submitting your revised manuscript:

We look forward to receiving your revised manuscript.

Kind regards,

James A.R. Marshall, BSc, PhD

Academic Editor

PLOS Computational Biology

Zhaolei Zhang

Section Editor

PLOS Computational Biology

Feilim Mac Gabhann

Editor-in-Chief

PLOS Computational Biology

Jason Papin

Editor-in-Chief

PLOS Computational Biology

**Journal Requirements:**

At this stage, the following Authors/Authors require contributions: Dieter Vanderelst, and Herbert Peremans. Please ensure that the full contributions of each author are acknowledged in the "Add/Edit/Remove Authors" section of our submission form.

5) We notice that your supplementary Figures are included in the manuscript file. Please remove them and upload them with the file type 'Supporting Information'. Please ensure that each Supporting Information file has a legend listed in the manuscript after the references list.

6) In the online submission form, you indicated that your data will be submitted to a repository upon acceptance. We strongly recommend all authors deposit their data before acceptance, as the process can be lengthy and hold up publication timelines. Please note that, though access restrictions are acceptable now, your entire minimal dataset will need to be made freely accessible if your manuscript is accepted for publication. This policy applies to all data except where public deposition would breach compliance with the protocol approved by your research ethics board. If you are unable to adhere to our open data policy, please kindly revise your statement to explain your reasoning and we will seek the editor's input on an exemption.

**Reviewers' comments:**

Reviewer's Responses to Questions

Reviewer #1: The submitted paper presents a simulation model that demonstrates how bats can move using echolocation calls in highly dense aggregations, by using the soundscape generated collectively rather than attempting to identify echoes of their own calls. The simulations are targeted at a particular species of bat (with excellent justification for this choice), and the authors have made a very good attempt at parameterizing the model to this species and comparing the results to known data about such bats. Overall I liked the approach of the study as it was built from a detailed and representative understanding of calling and hearing in bats (e.g. Figure 8), and the model tests the hypotheses very well, and the results are clear and convincingly support the stated findings.

The number of replicate simulations (runs) seems very small to me and there is no justification given for why 5 replicates per arena type is adequate. Similarly, the number of individuals in a run of the simulation (n=25) also seems very small compared to the numbers described earlier in the Introduction regarding the system which the simulations are representing. It would be good to see that the results are robust to higher numbers of individuals (if it is important to keep density constant, then the arena size can be scaled up appropriately). On a similar point, it would be good to see sensitivity analyses also for other chosen parameters such as the % in “we randomly placed the perched agents in 25% of the positions of the reflectors that make up the walls”.

More broadly, would the collective soundscape mechanism break down if different aspects of echolocation and hearing in the simulated bats was adjusted? This would allow the authors to test whether the calls and hearing have been tweaked by natural selection to optimize the collective soundscape effect; this wouldn’t be possible (or ethical) with the real system, especially in the wild and with 1000s of bats, so a simulation approach would be well suited to test this. My guess would be that the simulations are very computationally expensive to run and the number of replicates is limited, but this does need acknowledging in the text.

I found the presentation to be very good, especially the text which is easy to read and understand, and guides the reader through with just enough detail being given. There were some points regarding the figures and their legends:

The first sentence of figure 2’s legend needs to refer to the solid brown line. Is panel 2c referred to in the main text? Similarly, it wasn’t clear how the inset in 2b helped.

Figure 3’s legend needs an overall title explaining what the figure shows.

I didn’t think that the choice of colors for the line graphs was the best for clarity (lines of similar colors, beige background) and I wondered how these would look printed black and white or to those with common forms of color blindness.

On page 10, references 10, 29 to support “akin to the mechanisms underlying murmurations in birds [e.g., 10, 29]” are not ideal, as murmurations have been directly studied in bird flocks – see the influential studies by Andrea Cavagna and colleagues. These would be much more relevant references to cite here.

Christos Ioannou

University of Bristol

Reviewer #2: Vanderelst & Peremans - Plos Comp Biol

Data and Code Availability

Code documentation insufficient

~~~~~~~~~~~~~~~~~~~~

The reviewer appreciates the authors' broad documentation at the Zenodo level.

* In the actual 'code' folder however, the README is empty.

The following two comments are observations that the reviewer suggests the authors to consider making changes to - while being aware that Plos Comp Biol has no concrete guidelines for code conventions/quality. I suspect these suggestions are unlikely to change the scientific results, however - the comments are made at improving the quality of the methods from the perspective of (re)usability.

* The overview data on the Zenodo page has some broad information. This information can also be copied into the README file in the 'code' folder - with more user-relevant instructions.

* Aside from the name of the .py file, there is little information on what each script actually does. A short docstring description at the top of the script is needed to understand what the script sets out to do in a bit more detail.

* Function-level documentation is generally missing across modules. The authors may wish to use one of the three (Google/Numpy/Sphinx) styles.

Installation environment information missing

~~~~~~~~~~~~~~~~~~~~~~~~~~~~

For full replicability, the authors need to provide:

a) the Python version + package versions used to run the code. Alternately, a virtual environment or conda environment file with all the relevant information would also suffice.

b) Additional documentation on how to actually run the code, and the steps in replicating the results can also be included again in the README.

Incomplete code + data uploaded

~~~~~~~~~~~~~~~~~~~~~

In addition, there seem to be some missing modules/functions, included hard-coded paths to files that are not in the 'code' folder. I only tried three scripts, and already got the following errors due to missing modules, and leave the troubleshooting of the other scripts for the authors to follow up.

* 'SCRIPT_CaveEnsonifications.py' - 'ModuleNotFoundError: No module named 'acoustics''

* 'SCRIPT_CaveEnsonifications.py' - the pyBAT module is missing.

* 'SCRIPT_GetPingerZero.py' - 'ModuleNotFoundError: No module named 'acoustics'

Here I would suggest trying to run all the scripts with a clean-install, either using a different computer, or a new virtual environment to see how truly replicable the uploaded code is.

Section 1

---------

* "It is hypothesized that this helps avoid jamming and improves the selection of relevant echoes [reviewed by 27]." This statement is too brief to give full context of the literature. Also, to my understanding, ref 27 actually posits an opposite view. Suggest adding the original references instead of citing a secondary reference.

* "A surprising alternative way" - it is important to consider that this paper specifically studies 2 bats at a time. The first question is if the 'going silent' mechanism really scales beyond a small group-size, which it may not. More importantly, considering these are pairs of bats trained to fly & likely housed together, it is also important to consider there may be social factors contributing to the silences that do not occur in the swarming contexts discussed just before (e.g. see Wright et al. 2014, Curr Biol). Would consider not including this sentence/citation.

Section 2

---------

* 'We created simulated arenas...' - what was the dimension of the arena. The density of points is mentioned, but not the actual size. While the actual size is mentioned later, it helps to put the ~5m width here, and then refer to multiple variants modelled later.

* 'We did this to simplify the simulations, but also because bats following a flight path have been observed to maintain a constant flight height', also cite Kloerpper & Bentley 2017, Anim Behav - where they show stable flight heights during emergence.

* '...directionality of the bat Phillostomus discolor ’s emission and hearing as simulated by Vanderelst et al. ' fix typo in Phillostomus  Phyllostomus

* '...directionality of the bat Phillostomus discolor ’s emission and hearing as simulated by Vanderelst et al. ' - this is an odd choice considering Phyllostomids are known to be particularly directional echolocators. This assumption will 'reduce' the intensity of the problem as the typical non-phyllostomid bat has an omni-directional call emission.

* "This call rate also corresponds to the lowest rate observed by Lin et al. [25] for groups of Miniopterus fuliginosus leaving a cave" ...- though the methodology of cross-correlating a template call in reverberant audio makes one wonder about the actual call rates, which are much lower - which needs to be considered. Moreover, the likelihood of such a high call rate being sustained over a few seconds - a minute also seems low.

Figure 1

--------

* 'b)The color of the dot shows which type of acoustic input was the loudest at each plotted position. We also plotted the path of a single selected agent.' - The line corresponds to the flight path one agent, while the dots correspond to the loudest sound type heard by the agent along the flight path. If this is the case, then how come the agent flying into the right-most flight path can hear loudest sounds that are from the left branch of the cave system?

Results

-------

* Wrong hyperlinks - Fig. S1, S2 link to Main Figures 1 & 2!! I'm guessing the authors refer to the Figures in the Appendix?

Discussion

----------

* 'As such, our results could be interpreted as expanding the results of Mazar and Yovel [27] by further reducing the challenge jamming poses to swarming bats' - this is really a semantic point. In some sense, the authors are not 'expanding', but actually shifting the paradigm of investigation here. 'Jamming' is a problem only if we consider a signal and a 'jammer'. The authors here have neatly shown that both are informative in some sense!

* '..than the bats filmed by Weesner et al....' this 0.47 m matches the ~0.5 m also reported by Theriault et al. 2010 ('reconstruction & analysis of 3d trajectories of Brazilian Free-tailed bats in flight') - which the authors may consider citing in earlier parts of the manuscript too.

General comments

Some minor points that the authors may wish to clarify and/or discuss:

* Having read the methods, it seems like the simulations assume synchronised call emission from all agents at all times. Is this the case? If yes, then the authors need to make it explicit in the methods - and also discuss how this might actually make the task easier/harder for individual agents, vs. having unsynchronised call emissions. As shown in the cocktail party literature (and more specifically by Beleyur & Goerlitz 2019) unsynchronised calling across agents allows occasional auditory 'glimpses' of echoes too every now and then.

* The call rate of ~35 Hz seems very high esp for a bat that needs to fly for a few seconds to a minute maintaining that rate. The authors may consider discussing this choice. If a lower call rate had been used - perhaps the agents may have a harder time establishing the collective soundscape as their 'update rate' in come sense is lower?

* This may be my own misunderstanding of the methods - but it seemed like the agents take in the entire 30 ms audio and process it to generate their movement decisions. Whether biological echolocators take the entire interpulse-interval aside, this also inflates the amount of sensory information agents actually have access to. In contrast, when agents only consider the first X milliseconds (as the authors themselves have implemented in a previous paper) - this limits the amount of sensory information, and seems more realistic.

**Have the authors made all data and (if applicable) computational code underlying the findings in their manuscript fully available?**

Reviewer #1: Yes

Reviewer #2: **No: **Some modules are missing, and suspect that the files pointing to some hard-coded paths are also missing?

PLOS authors have the option to publish the peer review history of their article (what does this mean?). If published, this will include your full peer review and any attached files.

Reviewer #1: **Yes: **Christos C Ioannou

Reviewer #2: No

**Figure resubmission:**
---

## [Decision Letter · Decision Letter 1]

28 Feb 2025

PCOMPBIOL-D-24-01732R1

How swarming bats can use the collective soundscape for obstacle avoidance

PLOS Computational Biology

Dear Dr. Vanderelst,

Thank you for submitting your manuscript to PLOS Computational Biology. After careful consideration, we feel that it has merit but does not fully meet PLOS Computational Biology's publication criteria as it currently stands. Therefore, we invite you to submit a revised version of the manuscript that addresses the points raised during the review process.

Please submit your revised manuscript within 30 days Apr 30 2025 11:59PM. If you will need more time than this to complete your revisions, please reply to this message or contact the journal office at ploscompbiol@plos.org. Please include the following items when submitting your revised manuscript:

We look forward to receiving your revised manuscript.

Kind regards,

James A.R. Marshall, BSc, PhD

Academic Editor

PLOS Computational Biology

Zhaolei Zhang

Section Editor

PLOS Computational Biology

**Additional Editor Comments:**

I agree with reviewer 1 that the resubmission still does not address some important aspects of their review. As you will know, in computational simulation both sensitivity and robustness of results is of great importance. The reviewer's questions pertain to both of these aspects of computational science. However, while there is some justification of the choices made, the specific questions of the adequacy of the simulation numbers to support the conclusions, as well as the sensitivity to parameter variation, are not adequately addressed in the resubmission and the response. Please ensure these final points are engaged with fully.

**Journal Requirements:**

We have noticed that you have uploaded Supporting Information files, but you have not included a list of legends. Please add a full list of legends for your Supporting Information files after the references list.

**Reviewers' comments:**

Reviewer's Responses to Questions

**Comments to the Authors:**

Reviewer #1: The authors have addressed most of the issues I raised in the first round of review.

They have addressed the number of individual bats simulated in each run of the simulation, but not

“The number of replicate simulations (runs) seems very small to me and there

is no justification given for why 5 replicates per arena type is adequate.”

or

“On a similar point, it would be good to see sensitivity analyses also for other chosen parameters such as the % in “we randomly placed the perched agents in 25% of the positions of the reflectors that make up the walls”.”

in the response to reviewers’ comments. These points seem to limit the reliability of the results.

Also hasn’t been addressed is “More broadly, would the collective soundscape mechanism break down if different aspects of echolocation and hearing in the simulated bats was adjusted?”; the idea here is that within a simulation model, manipulation of the sensory systems of the agents is possible, and the impact of this on the collective level can be investigated. Currently its not obvious what aspects of the bat’s sensory traits are important in driving the observed behaviour.

Reviewer #2: * I see the authors have taken the time and made clarifications in multiple points, including figure legends and adding new figures - all of which greatly help the clarity of the paper's very exciting research direction.

* Slightly pedantic point, but needs to corrected for the record - a typo has remained from the previous version: 'Phillostomus discolor' -> 'Phyllostomus discolor'

* Appreciate the comparison on hearing and call directionality - despite the dearth of data on the studied species, these figures help establish a comparison of T. brasiliensis with other species. More broadly speaking of course, these figures help establish a general mechanism for any bat species that emerges at high densities.

Code

----

* This reviewer greatly appreciates the authors efforts at making the code more user-friendly and uploading it on the OSF platform

* One small suggestion would be to make a requirements.txt file also - this is an operating-system agnostic way to listing the necessary packages. The conda environment file provided is appreciated - while the conda environment file is detailed the one associated disadvantage is its OS-specific nature. At this point of time I'm not entirely sure what other options are available however!

* At least one hard-coded path (have not been able to check for more due to time constraints) in the code remains (e.g. could not run 'SCRIPT_Swarm.py'  due to line 16 in ReadArena.py with a hard-coded path) - which is an issue in validating the replicability of the code base for an external user/reviewer. I acknowledge once again that reviewer-replication is not a requirement from the Plos Comp Biol side, but do urge the authors to check for other such hard-coded paths, especially by running the current code-base on a different computer.

**Have the authors made all data and (if applicable) computational code underlying the findings in their manuscript fully available?**

Reviewer #1: Yes

Reviewer #2: Yes

PLOS authors have the option to publish the peer review history of their article (what does this mean?). If published, this will include your full peer review and any attached files.

Reviewer #1: No

Reviewer #2: No

**Figure resubmission:**
---

## [Editor Report · Decision Letter 2]

31 Mar 2025

Dear Dr. Vanderelst,

We are pleased to inform you that your manuscript 'How swarming bats can use the collective soundscape for obstacle avoidance' has been provisionally accepted for publication in PLOS Computational Biology.

Best regards,

James A.R. Marshall, BSc, PhD

Academic Editor

PLOS Computational Biology

Zhaolei Zhang

Section Editor

PLOS Computational Biology

Thank you for addressing the outstanding questions raised by the reviewers. I am happy to accept this revised manuscript.

---

## [Editor Report · Acceptance letter]

PCOMPBIOL-D-24-01732R2

How swarming bats can use the collective soundscape for obstacle avoidance

Dear Dr Vanderelst,

I am pleased to inform you that your manuscript has been formally accepted for publication in PLOS Computational Biology. Your manuscript is now with our production department and you will be notified of the publication date in due course.

With kind regards,

Anita Estes
